# Counterexample Guided RL Policy Refinement Using Bayesian Optimization

**Briti Gangopadhyay**[*]
Department of Computer Science
Indian Institute of Technology Kharagpur
`briti_gangopadhyay@iitkgp.ac.in`

**Pallab Dasgupta**
Department of Computer Science
Indian Institute of Technology Kharagpur
`pallab@cse.iitkgp.ac.in`

## Abstract

Constructing Reinforcement Learning (RL) policies that adhere to safety require-
ments is an emerging field of study. RL agents learn via trial and error with an
objective to optimize a reward signal. Often policies that are designed to accumu-
late rewards do not satisfy safety specifications. We present a methodology for
counterexample guided refinement of a trained RL policy against a given safety
specification. Our approach has two main components. The first component is an
approach to discover failure trajectories using Bayesian Optimization over multiple
parameters of uncertainty from a policy learnt in a model-free setting. The second
component selectively modifies the failure points of the policy using gradient-based
updates. The approach has been tested on several RL environments, and we demon-
strate that the policy can be made to respect the safety specifications through such
targeted changes.

## 1 Introduction

Classical Reinforcement Learning (RL) is designed to perceive the environment and act towards
maximizing a long term reward. These algorithms have shown superhuman performances in games
like GO [10], motivating their use in safety-critical domains like autonomous driving [23, 20, 16].
These policies are mostly learnt in simulation environments such as TORCS [22] or CARLA [14]
for accelerated learning. Therefore they are susceptible to perform poorly in the real world where
domain uncertainty arises due to noisy data. Also, the choice of a RL policy is determined by the
design of the reward signal, which if not constructed properly, may lead the agent towards unintended
or even harmful behaviour. Hence the need for constructing policies that account for risks arising due
to uncertainty and do not violate safety specifications.

Safe RL aims to learn policies that maximize the expectation of the return while respecting safety
specifications [17]. The existing literature on safe RL is broadly divided into three schools of
thought namely, 1) Transforming the optimization criteria by factoring in costs derived from safety
constraints [1], 2) Using external knowledge in the form of a teacher or a safety shield to replace
unsafe actions chosen by the policy during exploration / deployment [2], and 3) Establishing the
existence of an inductive safety invariant such as a Lyapunov function [11] or a control barrier
function [25]. Establishing such an invariant is often intractable for Neural Network controllers that
use many parameters and work on high-dimensional state spaces.

In this work, we propose a *counterexample guided policy refinement* method for verifying and
correcting a policy that has already been optimized and calibrated. Failure traces may occur in
such a policy due to the following reasons: a) Failure trajectories are extremely rare and do not
contribute significantly enough to reduce the expected reward, b) Uncertainty in the environment

---

[*]Code is available online at https://github.com/britig/policy-refinement-bo

35th Conference on Neural Information Processing Systems (NeurIPS 2021).

which had not been accounted for during policy training. Our aim is to progressively make the system safer by finding and eliminating failure traces with targeted policy updates. Counterexample guided refinement techniques have been well studied in the formal methods community for program verification [9] and reachability analysis of hybrid systems [3], but not so far in the context of RL.

We study the proposed methodology on environments that work over continuous space and continuous or discrete actions. The proposed methodology is divided into two steps:

1. Given a policy $\pi_{old}$, learnt for optimizing reward in a given environment, we test it against parameters with uncertainties and a set of objective functions $\varphi$ derived from the negation of the given safety criteria. Multiple counter-example traces or failure trajectories are uncovered from $\pi_{old}$ using Bayesian Optimization.

2. Using the failure trajectories we selectively do a gradient update on $\pi_{old}$ to construct a new policy $\pi_{new}$, that excludes the counterexample traces under the given domain uncertainties. We specifically work with a policy gradient RL algorithm, Proximal Policy Optimization (PPO), which directly optimizes the learnt policy using gradient updates.

These steps may be repeated multiple times. We show that such targeted updates ensure that $\pi_{new}$ is safer than $\pi_{old}$, but not significantly different in terms of reward.

The paper is organized as follows. Section 2 outlines related work, Section 3 presents the methodology to uncover failure trajectories, Section 4 presents the methodology of policy refinement, Section 5 presents case studies on several RL environments, and Section 6 provides concluding remarks.

## 2 Related Work

The approach proposed in this paper uses Bayesian Optimization for finding adversarial tests cases in terms of trajectories for the RL policy leading to failure of the safety specification. The failures are then used to make the policy safer.

Bayesian Optimization (BO) [28], is a machine learning technique for finding the global maxima/minima of an unknown function based on stochastic evaluations. BO has been extensively used in combinatorial optimization [4], hyper-parameter optimization for neural networks [29] and optimization of parameters in robotics. In recent literature, BO has been successfully applied for generation of adversarial counterexamples for complex controllers [13, 18] including identification of test cases for autonomous driving systems [15], and coverage of faults in analog and mixed circuits [19]. Traditional Bayesian Optimization is limited to finding a single global minima/maxima which can lead to the discovery of only one counterexample. Hence, extensions that find multiple test cases has been proposed on top of traditional BO [15, 19]. In this work, we closely follow the formulations and extensions of [15].

Safe RL is an area of emerging significance buoyed by the proliferation of RL in safety-critical control systems. A survey of RL safety problems and taxonomies can be found in [17]. The goal of safe RL is to create agents whose behaviour is aligned with the user's intentions. Achieving this via reward shaping [21] causes a trade-off between a reward enacting task-solving behaviour and a penalty for non-avoidance of hazards. Less penalty promotes unsafe behaviour and high penalty dilutes the objective goal of the RL.

A natural way to incorporate safety specification in RL is by imposing constraints. A substantial body of work exists in this direction [24, 12, 7, 8, 1]. However, there is no obvious guarantee that the framework of constrained RL solves the agent alignment problem. Constrained RL requires the specification cost functions for the constraints, along with the reward function, with susceptibility towards design errors [26]. In the proposed method we neither change the reward function nor introduce constraint costs. Our policy corrections are affected by changing the policy gradient based on violations of the safety specifications.

Another school of work uses Preference-Based RL, where the agent learns directly from an expert's preferences of trajectories instead of a hand-designed numeric reward. Preference-Based RL suffers from the temporal credit assignment problem, that is, determining which states or actions contribute to the encountered preferences [31]. Counterexample guided refinement has been studied in Apprentice Learning (AL) using PCTL model checking [32]. Our approach is applicable to a wider range of applications since it can handle larger state spaces than the model checking approach from [32].

The authors in [6] do a counterexample based improvement on RNN-based strategy on a POMDP using Linear Programming. Model-checking is performed on an RNN strategy induced MDP and the strategy is refined using Linear Programming based on counterexamples. The application of their work is limited to environments with discrete state space. In their setting, the POMDP model is known apriori, which is not true in our setting.

## 3   Finding Failure Traces using Bayesian Optimization

This section outlines our approach for using Bayesian Optimization to look for failure traces in a trained RL policy, $\pi_{old}$, with respect to a given safety specification. BO is a machine learning approach formulated as:

$$\underset{p \in P}{\operatorname{argmin}} f(p)$$

where $f : \mathbb{R}^d \to \mathbb{R}$ is continuous but lacks special structure and $p \in P$ where $P$ is a set of function parameters. BO is used to optimize derivative free, expensive-to-evaluate black box functions. In its core BO uses Gaussian Process (GP) models to estimate a prior over the function $f$. The function value, $f(p)$, is treated as a random variable by a Gaussian Process, which enables the modelling of a finite number of observations using a joint Gaussian distribution. The distribution initially has an assumed mean, $m(p)$, at 0 and the kernel, $\kappa(p_i, p_j)$, is a squared exponential function of the parameters $p$, $e^{-(|p_i - p_j|)^2}$. For every unobserved value $p_*$, the mean, $m$, and variance, $\sigma^2$ can be estimated using the following equations [28]:

$$m(p_*|p) \quad = \quad m(p_*) + K_*^T K^{-1}(f(p) - m(p)) \tag{1}$$

$$\sigma^2(p_*|p) \quad = \quad K_{**} - K_*^T K^{-1} K_* \tag{2}$$

where $K = \kappa(p, p)$, $K_* = \kappa(p, p_*)$, and $K_{**} = \kappa(p_*, p_*)$. An acquisition function is then used to sample the estimated prior, along with assumed mean and variance, for unobserved points to derive a posterior distribution. The acquisition function chosen in this paper is *Expected Improvement*, which represents the quantum of improvement while exploring and exploiting the function to find a global minimum. If $f^*(p)$ is the minimal value of $f(p)$ observed so far and $f'(p)$ is the new evaluation, then the expected improvement utility, *EI*, is given by:

$$EI(p) = E[\max(0, (f^*(p) - f'(p)))]$$

where, $E$ indicates the expectation taken under the posterior distribution, given evaluations of $f$. The expected improvement algorithm then chooses to evaluate the function at the point having the largest expected improvement, namely:

$$p_{n+1} = \operatorname{argmax} EI(p)$$

The prior model improves through repeated sampling using the acquisition function until the improvement becomes insignificant (indicative of a global minima) or the computation budget is exhausted.

We now explain the use of BO in our framework. The following example elucidates the problem.

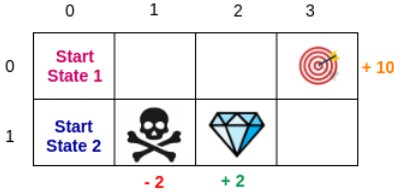

Figure 1: A toy grid environment

**Example 3.1** *The toy grid-world of Fig. 1 has two start locations at (0,0) and (1,0), and a single goal location at (0,3). The agent has two actions, "left" for moving left, and "up" for going up. The reward is indicated beside some of the locations, and is -1 for all other locations. Suppose the safety requirement is that the agent should never reach a location having a reward less than -1, which is modeled by the predicate:* $\neg(reward(state) + 1 < 0)$. *In the absence of this safety requirement, the following trajectories are optimal in terms of total reward:*

*Trajectory-1:*   $(0,0) \xrightarrow{left} (0,1) \xrightarrow{left} (0,2) \xrightarrow{left} (0,3)$

*Trajectory-2:*   $(1,0) \xrightarrow{left} (1,1) \xrightarrow{left} (1,2) \xrightarrow{left} (1,3) \xrightarrow{up} (0,3)$

*Trajectory-2 violates the safety specification at (1,1), and is therefore a failure trajectory. BO helps us to find such failure traces automatically, by intelligently testing the RL policy with adversarial tests derived from the safety specification.*

**Algorithm 1:** Finding Failure Trajectories

---

**Input:** Policy $\pi_{old}$, Parameter Bounds $P$, Objective functions $\varphi$, Agent $A$
**Function** `identify_failures`($\pi_{old}, P, \varphi, A$):

> $i \leftarrow 1$
> **while** $i < max\_budget$ **do**
>> Sample parameters $p_i \in [P_{low}, P_{high}]$ based on GP models
>> $j \leftarrow 1$
>> $o_j \leftarrow p_i$
>> **while** *not done* **do**
>>> $a_j \leftarrow \pi_{old}(o_j)$
>>> $o_{j+1} \leftarrow A.step(a_j)$
>>> $\xi_i \leftarrow \xi_i \cup (o_j, a_j)$
>>> $o_j \leftarrow o_{j+1}$
>>> $j \leftarrow j + 1$
>>
>> **end**
>> $\varphi_{val_i} \leftarrow argmin_{\xi_i} \ (\mu_1(\xi_i)) \ldots (\mu_n(\xi_i))$
>> Update each GP model with measurements $(p_i, \mu_k(p_i))$
>> $\xi_f \leftarrow \xi_f \cup (\xi_i \text{ with } \varphi_{val_i} < 0)$
>> remove $p_i$ from search space
>> $i \leftarrow i + 1$
>
> **end**
> **return** $\xi_f$

**Comments:**
$\varphi$ : Set of objective functions corresponding to predicates in negation of safety criteria.
$\xi_i$: A sequence of states $o_t$, actions $a_t$ over time $t = 0, 1, \ldots$
$\varphi_{val_i}$: Evaluation of the objective functions in $\varphi$ for the $i^{th}$ trajectory.

---

In the proposed methodology we sample the given policy, $\pi_{old}$, for failure trajectories using Algorithm 1 which aims to minimize a set of objective functions, $\varphi(p)$, derived from the specification over the given set of parameter bounds $P$. $\varphi(p)$, is a manifestation of the negation of the safety criteria in terms of a set of objective functions that can be minimized. The function set $\varphi$ derived from the safety criteria may involve negation, conjunction, and disjunction of multiple predicates over the variables in the system i.e., $\varphi := \mu \mid \neg\mu \mid \mu_1 \wedge \mu_2 \mid \mu_1 \vee \mu_2$. It is assumed that each predicate function $\mu_k$ is a smooth and continuous function over trajectories $\xi$. Violation of any $\mu_k \in \varphi$ forms the basis of violation for the overall safety criteria. Each such component, $\mu_k$ is modeled as a separate GP in our methodology. This method is able to uncover more counterexamples than modeling $\varphi$ with a single GP as demonstrated by [18]. For cases where a set of predicates need to be jointly optimized as shown in example 3.2 we normalize the variables. In each iteration, the parameter values selected by BO are evaluated against each $\mu_k$ and the evaluated value is used by the acquisition function to push each $\mu_k$ towards its minimum. Each trajectory, containing sampled parameters $p_i$, that violates any $\mu_k$ is added in the failure trajectory set. This process continues until the entire sample space is covered i.e all the possible states that can cause failure is removed from the search space, or the time budget is exhausted. We use the extension of BO provided in [15], that removes the failure area enclosed by an already explored trajectory from the search space, to uncover multiple failure trajectories.

**Example 3.2** *A safety specification for lunar lander trained using RL might be that the lander cannot be tilted at an angle while being close to the ground. The lander coordinates are specified by $(l_x, l_y)$ and the angle is given by the variable $l_{angle}$. Concretely, let us assume $0 \leq l_y \leq 10$ and $-1 \leq l_{angle} \leq 0$. Using predicate logic we can represent this safety specification as $l_y < 5 \rightarrow l_{angle} \geq -0.5$, which in words says that if the lander is at a vertical distance of less than 5m from the ground then the angle of the lander is greater than -0.5. The negation of this specification can be rewritten as $\varphi : (l_y < 5) \wedge (l_{angle} < -0.5)$. There are two predicates in the specification $\mu_1 : l_y - 5 < 0$ and $\mu_2 : l_{angle} + 0.5 < 0$. This can be converted as an equivalent BO objective function as $min(\mu_1 + \mu_2)$. Intuitively, this means that if $\varphi < 0$ the safety specification is violated, for example one such evaluation of the function could be $l_y = 1$ and $l_{angle} = -0.8$. An example is shown in Figure 2b.*

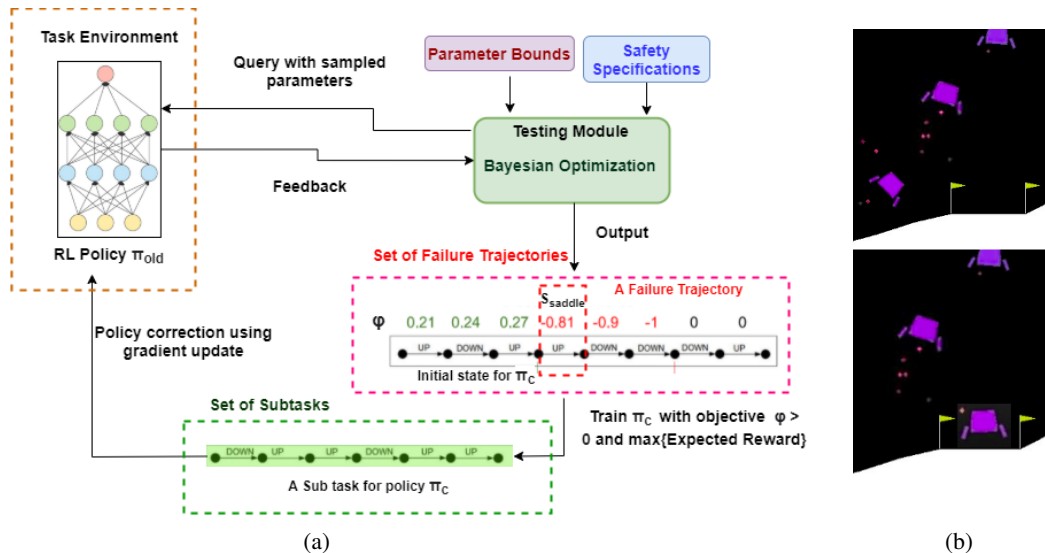

Figure 2: a) Overview of the policy refinement framework. The failure trajectories are obtained by executing BO on the original learnt policy. The failure trajectories are then used for learning a new sub-policy which acts as the reference for gradient updates when refining the original policy. b) An example of a failure trajectory and the corresponding corrected trajectory on Lunar Lander Environment.

# 4 Policy Refinement Methodology

Fig. 2a outlines the proposed counter-example guided policy refinement approach. In each iteration, the previous policy is tested using the approach proposed in Section 3 to find a set of failure trajectories. This section presents the proposed methodology for using these failure trajectories to refine the policy. We begin with a set of formal definitions.

**Definition 4.1 (Failure Trajectory)** *A failure trajectory, $\xi_{fi} = \{s_0, a_0, s_1, a_1, \ldots, s_t, a_t\}$, is a sequence of state-action pairs that contains one or more states where the safety specification is refuted i.e. $\exists s_i \in \xi_{fi} : \mu_k(s_i) < 0$ where $\mu_k \in \varphi$. $\xi_f$ denotes the set of failure trajectories.*

**Definition 4.2 (Saddle point of failure trajectory)** *We define saddle point as the first state on a failure trajectory where the prediction made by the actor network violates the safety specification.*

Following example 3.1, we want to change the policy for the grid-world from the sate before the saddle state (1,1) i.e., from (1,0) in Trajectory 2. The updated trajectory to be enforced in the policy network is $(1,0) \xrightarrow{up} (0,0)$ requiring minimum gradient update. Algorithm 2 formally outlines the proposed policy refinement methodology. For each failure trajectory $\xi_{fi} \in \xi_f$ obtained using Algorithm 1, we identify the saddle point, $s_{saddle_i}$. We use the states preceding the saddle state, $s_{saddle_{i_0}}$ where $i_0 < i$, in $\xi_{fi}$ as the initial states of a new sub-policy $\pi_c$ that learns the correct actions for all trajectories in $\xi_f$. The objective of $\pi_c$ is to satisfy the specification, namely to have $\varphi_{val} \geq 0$ on states of the new trajectory, while also maximizing the reward function. To incorporate a negative penalty for the violation of the safety specification we add the evaluation of the objective functions $\varphi_{val_i}$ with an importance of $\beta$ along with the reward of a trajectory $\xi_{fi}$ while training $\pi_c$.

We considered the possibility of refining the original policy $\pi_{old}$ by using $\pi_c$ as a *safety shield*, that is, to override the action of $\pi_{old}$ by the action of $\pi_c$ in selected states. In this approach, $\pi_c$ does not force $\pi_{old}$ to learn possibly more rewarding policies by exploring ways around the saddle points. In other words, forcible preemption of actions proposed by $\pi_{old}$ results in a policy which is distant from $\pi_{old}$ and potentially less rewarding. Moreover disruptive changes in an optimized policy is typically not welcome in any industrial setting.

Therefore we propose to refine $\pi_{old}$ using a *proximal policy gradient* approach, where we use $\pi_c$ to change the policy gradient near the saddle points in a way that $\pi_{old}$ adjusts itself locally to a new policy $\pi_{new}$. Over time, $\pi_{new}$ learns to avoid the failure trajectories through such local adjustments.

---
**Algorithm 2:** Policy Refinement Algorithm
---
**Input:** Policy $\pi_{old}$, Failure Trajectories $\xi_f$
**Function** `Correct_Policy`$(\pi_{old}, \xi_f)$:
    **for** *each $\xi_{fi}$ in $\xi_f$* **do**
        $R_i = R_i + \beta * \varphi_{val_i}(\xi_{fi})$
        Train $\pi_c$ using $(\xi_{fi}, s_{saddle_i}, R_i)$
    **end**
    **for** *each $\xi_{fi}$ in $\xi_f$* **do**
        Collect observation data $D_k$ for $\xi_{fi}$
        Collect new actions and log probabilities corresponding to $D_k$ from new sub-policy $\pi_c$
        Set Advantage Estimate $A_{t_k}$ to 1
        Update $\pi_{old}$ to $\pi_{new}$ by maximizing the PPO clip objective using $\pi_c$
    **end**
**Comments:**
$\beta$ : Importance value of the evaluated safety specification.
---

Since the adjustments are local, $\pi_{old}$ and $\pi_{new}$ are proximal and therefore (typically) not very different in terms of reward. In order to formalize this approach, we outline the policy gradient methodology, followed by our proposed intervention to use it in the above context.

Suppose $\langle \mathcal{S}, \mathcal{A}, P^a_{ss'}, \mathcal{R}, \gamma \rangle$ is a finite horizon discounted Markov Decision Process (MDP), where $\mathcal{S}$ is the set of states, $\mathcal{A}$ denotes the actions, and $P^a_{ss'}$ denotes the probability of transition from state $s \in \mathcal{S}$ to state $s' \in \mathcal{S}$ by taking action $a \in \mathcal{A}$. Further, $\mathcal{R}$ denotes the reward signal and $0 \leq \gamma \leq 1$ is the discount factor. The next state of the environment, $s_{t+1} \in S$, is determined stochastically according to a joint distribution depending on the state-action pair $(s_t, a_t)$.

The objective of any policy gradient RL algorithm [30] is to learn the best action to take in it's policy $\pi$, by fine-tuning a set of neural network parameters $\theta$. The policy, $\pi$, is defined by $P_r\{A_t = a | S_t = s, \theta_t = \theta\}$, namely the probability of taking action $a_t$ at state $s_t$ under parameters $\theta_t$. Popular policy gradient algorithms follow an actor-critic architecture where the critic estimates the value function by updating the value function parameters and the actor updates the policy parameters, $\theta$, for $\pi_\theta(a|s)$, in the direction suggested by the critic. We focus on a particular class of Policy Gradient Algorithms known as Proximal Policy Optimization (PPO) [27]. The core idea behind PPO is to use a *clipped surrogate objective* that improves the training stability by limiting the changes made to the policy at each step. The policy update objective function is given by

$$L^{clip}_\theta \quad = \quad \hat{\mathbb{E}}\left[ min\left( r_t(\theta), clip(r_t(\theta), 1 - \epsilon, 1 + \epsilon) \right) . \hat{\mathbb{A}}_t \right] \tag{3}$$

$$\text{where} \quad r_t(\theta) \quad = \quad \frac{\pi_\theta(a_t|s_t)}{\pi_{\theta_{old}}(a_t|s_t)} \tag{4}$$

Following the theory of importance sampling, $r_t(\theta)$ calculates the ratio between the probability of the action under the current policy $\pi_\theta(a_t|s_t)$, and the probability of the action under the previous policy $\pi_{\theta_{old}}(a_t|s_t)$, to trace the impact of the actions. To avoid drastic changes during policy update, the clipped surrogate function, $L^{clip}_\theta$, takes the minimum of the clipped ratio and the actual ratio multiplied with the advantage, $\mathbb{A}_t$.

In the proposed counter-example guided policy refinement approach, the machinery used for gradient update of the actor network, $\pi_{old}$, is the same as the gradient update in original PPO algorithm except that we intend to use $\pi_c$ to update the gradient. Towards this goal we introduce the following modifications:

1. We change the clipped objective ratio $r_t(\theta)$ as follows:

$$r_t(\theta) = \frac{\pi_{old}(a_t|s_t)}{\pi_c(a_t|s_t)} \tag{5}$$

where $\pi_c$ is the sub policy with correct actions for failure trajectory $\xi_{fi}$ and $\pi_{old}$ is the old policy that we want to refine to a new policy $\pi_{new}$.

2. We set the advantage factor $\mathbb{A}_t$ to be 1 for the following reason. During the correction process, the action predicted by $\pi_c$ given a failure trajectory forms the mean of the probability distribution

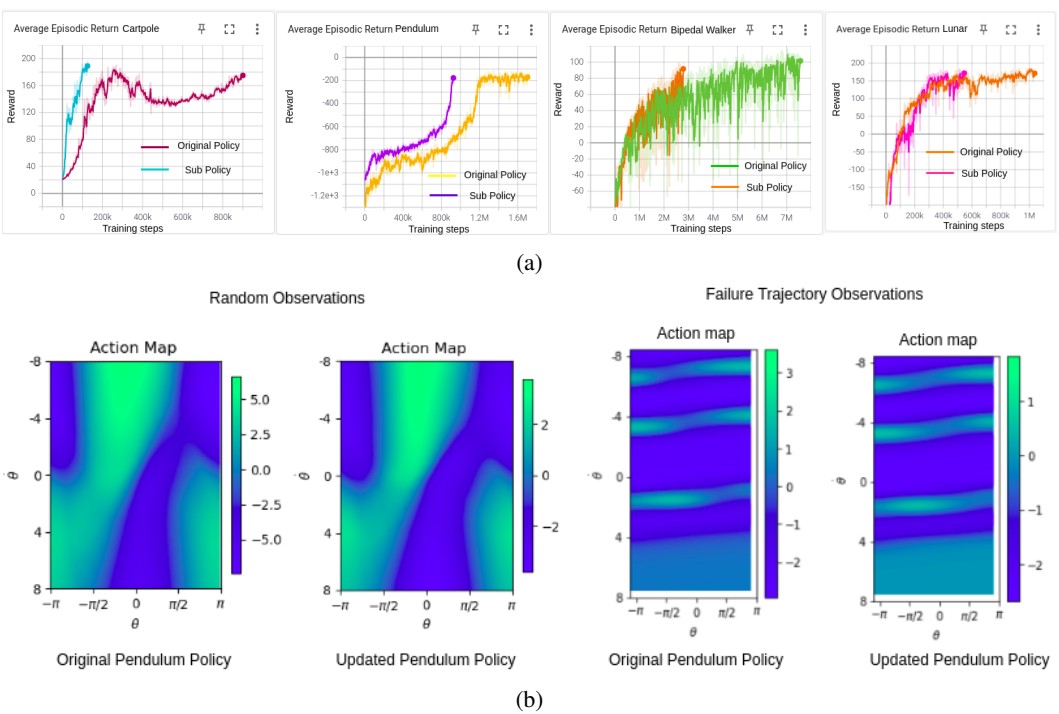

Figure 3: a) Reward plots for different environments showing training steps and the reward obtained by the sub-policy and the original policy b) Observation vs action heat-map for pendulum policy

from which the action is sampled, that is, $a_i \sim \mathcal{P}(\pi_c(o_i))$ where $o_i \in \xi_{fi}$ is an observed state in the trajectory. $\mathcal{P}$ can follow a Categorical distribution (for discrete actions) or a Multivariate Normal distribution (for continuous actions). We only consider the (observation,action) pairs that lead to the satisfaction of $\varphi$ over the entire trajectory while sampling from $\pi_c$. Since, these corrected trajectories are to be enforced into $\pi_{old}$ we set the advantage factor $\mathbb{A}_t$ to be 1.

Section 5 provides empirical evidence in support of the proposed methodology in terms of safety and reward. We also emphasize that the refined policy $\pi_{new}$ obtained using our method is minimally different from the old policy $\pi_{old}$, thereby ensuring minimal trade-off of reward in favor of safety. The following metric defines the distance between the two policies.

**Definition 4.3** *Variation Distance between two policies: Since we work with model free reinforcement learning, we define the variation distance between two policies in terms of their trajectories. We sample n random trajectories from both the policies along with the failure trajectories from policy $\pi_{old}$ and their corresponding updated trajectories from policy $\pi_{new}$. We want to measure the difference of states visited between the trajectories. The distance function $D_v$ is given as follows where $\xi_i, \xi_i'$ correspond to the trajectories of $\pi_{old}$ and $\pi_{new}$ respectively.:*

$$D_v(\pi_{old} \| \pi_{new}) = \frac{1}{n} \sum_{\xi_i, \xi_i' \in \xi} \sqrt{\sum_{s_i \in \xi_i, s_i' \in \xi_i'} |(s_i)_{\pi_{old}} - (s_i')_{\pi_{new}}|^2} \tag{6}$$

## 5 Empirical Studies

We test our framework on a set of environments from Open AI gym [5]. The environments, safety specifications, parameter bounds, number of failure trajectories and the mean variation distance, along with the standard deviation, between the original and updated policy is summarised in Table 1. Fig 3a shows the plots of rewards obtained by the original policy and the sub-policy in each environment. The sub-policies have a lower training time due to less number of trajectories to train on. The experiments were run on a machine with AMD Ryzen 4600h 6 core processor and GeForce GTX 1660 Graphics unit. We set $max\_budget = 200$ for each iteration of BO. The mean and

Table 1: Description of assertions tested on PPO policies along with the parameters bounds for BO query space. We also report the number of failure trajectories uncovered and corrected for each policy and the variation distance between the updated policy and the original policy. In our experiments n=1000

| Environment | Safety Criteria | Parameter Bounds | Failures | Distance |
|---|---|---|---|---|
| Cart-pole-v0 | 1. -2.4 < position < 2.4 
 2. -2.0 < momentum < 2.0 
 3. angle > 0.2 | State : [(-0.05, 0.05)] * 4 
 Mass : (0.05, 0.15) 
 Length of pole : (0.4, 0.6) 
 force magnitude: (8.00, 12.00) | 174.4 
 ± 
 0.51 | 1.255 
 ± 
 0.195 |
| Pendulum-v0 | 1. Reward > -300 | $\theta$ : $(-\pi,\pi)$ 
 $\dot{\theta}$: (0,1) 
 speed: (-1,1) | 80.1 
 ± 
 1.85 | 10.866 
 ± 
 1.379 |
| BipedalWalker-v3 | 1. Hull Position > 0 
 2. -0.8 < Hull Angle < 2 | Hull angle : $(0, 2 * \pi)$ 
 Velocity x: (-1,1) 
 Velocity y: (-1,1) | 40.6 
 ± 
 4.08 | 11.189 
 ± 
 1.375 |
| LunarLander Continuous-v2 | 1. -0.4 < Landing $Position_x$ < 0.4 
 2. $Pos_y$ < 0.1 → (angle > -1 ∨ angle < 1) 
 3. Reward > 0 | $x_\delta$ : (0,10) 
 $y_\delta$: (0,20) 
 $velx_\delta$: (0,3) 
 $vely_\delta$: (0,3) | 40.85 
 ± 
 5.14 | 2.215 
 ± 
 0.282 |

variance of the number of failure trajectories uncovered is reported over 20 BO evaluations.

***Cart-pole Environment***: In this environment the pole on top of the cart is controlled by applying a force of +1 or -1 to the cart and the goal is to prevent the pole from falling over. The original PPO policy is tested against the following assertions:

1. The cart should not go beyond -2.4 (extreme left) or +2.4 (extreme right).

2. The cart maintains a momentum between -2.0 and 2.0

3. The angle made by the pole should be greater than 0.2 with respect to the rest position

***Pendulum Environment***: The pendulum-v0 environment models an inverted pendulum swing-up problem. The pendulum starts in a random position, and the goal is to swing it up so that it stays upright. We wish to ensure the pendulum does not swing twice on its pivot to stabilize i.e., the reward obtained is not less than -300. There are three sources of uncertainty as described in Table 1. Figure 3b shows the observation vs action map for a set of random observations and for observations of a failure trajectory. We can observe that actions with high values have reduced in the updated policy. However, the positive and negative action distribution over the observations remain the same.

***Bipedal Walker***: The goal in the bipedal walker environment is to navigate a two legged bot successfully to the end of a terrain. This is a challenging environment to test having 24 variables as a part of the state space and 4 action variables for different joints of the bot all of them taking their values from continuous domains. The robot falls on the ground when the hull angle goes below -0.8 (falling towards the left) or the angle is above 2 (falling towards the right). Hence, we want to find trajectories where hull angle < -0.8 or hull angle > 2 which forms predicates for our objective function. Also, the episodes terminates if the walker does not move at all (hull position < 0).

***Lunar Lander***: The task in this environment is to land a lunar lander vehicle smoothly between the landing flags at coordinate (0,0). We consider four sources of disturbances for the system. This can occur in the real world for a variety of reason such as noisy senor readings. The disturbance $(x_\delta, y_\delta)$ are added to the initial (x,y) coordinates where $x_\delta \in (0, 10)$ and $y_\delta \in (0, 20)$. Also disturbance $(velx_\delta, vely_\delta)$ are added to the initial horizontal and vertical linear velocity where $velx_\delta, vely_\delta \in (0, 3)$. Under these disturbances we want to the policy to satisfy the following assertions.

1. The lander should not land beyond a certain distance from the flag.

2. If the lander is close to the ground then it should not be tilted beyond a certain angle.

Table 2: Comparison with Baselines

| Environment | Policy A | Policy B | Policy C | Policy D |
|---|---|---|---|---|
| Cart-pole | Failures: 179
Training Steps: 900K | Failures: 52
Training Steps: 150K | Failures: 0
Training Steps: 1M | Failures: 0
Training Steps: 150K
+ 80K (Update) |
| Pendulum | Failures: 89
Training Steps: 1.6M | Failures: 102
Training Steps: 850K | Failures: 0
Training Steps: 1.8M | Failures: 89
Training Steps: 850K
+ 20K (Update) |
| BipedalWalker | Failures: 45
Training Steps: 7.5M | Failures: 145
Training Steps: 2.8M | Failures: 41
Training Steps: 8M | Failures: 0
Training Steps: 2.8M
+ 20K (Update) |
| LunarLander Continuous | Failures: 42
Training Steps: 1.1M | Failures: 18
Training Steps: 400K | Failures: 5
Training Steps: 1.2M | Failures: 0
Training Steps: 400K
+ 20K (Update) |

For the first assertion the failure trajectories have a reward greater than 200 as the rover lands successfully but lands beyond the allowed distance from the flags which violates the safety specification exhibiting that even trajectories that have positive rewards can violate safety specifications.

As exhibited by Table 1 BO is able to find failure trajectories, as many as 175 for cart-pole environment, from a tuned policy which had converged to a high reward. As displayed by Figure 3b and the variation distance reported in Table 1 the updation does not lead to a policy significantly different from the original policy in terms of optimization goal. We compare the following baseline PPO policies with the refined PPO policy:

(A) PPO policy trained from scratch with negative penalty for property violation, (B) PPO policy trained from scratch with only counterexample traces and negative penalty after one iteration of testing with BO same as $\pi_c$, (C) PPO policy trained from scratch with original training traces, counterexample traces and negative penalty after testing with one iteration of BO, and (D) The refined policy $\pi_{new}$. The observations are reported in Table 2. Even though the Policy A was penalized with respect to the satisfaction of the safety property, the variable variations which occurred during deployment were not known during training. Hence, Policy A has high number of counterexamples. Policy B has counterexamples that had not occurred in the original policy as it was only trained on counterexample traces and did not retain the optimized behaviour of the original policy. In some cases, Policy C has fewer counterexamples than Policy A for the same tests but the training time is substantially higher than our policy refinement method. There are also cases (bipedal walker) where Policy C has a high number of counterexamples even after the inclusion of counterexample traces and negative penalty, as the updates are not targeted. The refined policy by our methodology did not report any counterexamples for the same property even after 200 BO iterations and works with much fewer training and update steps.

## 6 Conclusions and Discussion

Adversarial testing of deep neural networks is critical for deployment of learned systems in safety critical applications. In the context of deep reinforcement learning, the problem is more complex as adversarial testing needs to find *failure trajectories* by searching over the space of states, actions, and parameters. The methodology presented in this paper targets the search for failure trajectories by posing it as a Bayesian Optimization problem, and then uses a policy gradient update mechanism to locally retrain the policy to avoid unsafe states. We believe this is the first work on counter-example guided policy gradient update, and the method appears promising based on empirical evidence.
A limitation of our method is in providing formally guaranteed safety which can be provided via formal verification methods or existence of lyapunov based policies on model based techniques where the MDP is known apriori. The method in its current form is also incapable of handling temporal safety specifications. As we find counter-examples using Gaussian Processes to estimate a prior of the safety specification function, the function must be a smooth and continuous function of the trajectory.

# 7 Broader Impact

Our paper introduces a methodology for testing and correcting RL policies that have already been optimized and calibrated. RL has the potential to bring significant benefits for society, beyond the boundaries of solving tasks in simulation environments. In many safety critical domains, it is of paramount importance to develop RL approaches that respect safety constraints during deployment. RL policies need to adapt and accommodate safety specifications of an ever changing environment without going through and entire training cycle. We believe that our work contributes to this quest, potentially bringing RL closer to high-stakes real-world deployment. Of course, any technology – especially one as general as RL – has the potential of misuse, and we refrain from speculating further.

## Acknowledgments and Disclosure of Funding

The authors would like to thank Department of Science and Technology (DST), Government of India and TCS Research Scholarship for partially supporting this work.

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
