# Supplement for Counterexample Guided RL Policy Refinement Using Bayesian Optimization

**Briti Gangopadhyay**[*]
Department of Computer Science
Indian Institute of Technology Kharagpur
briti_gangopadhyay@iitkgp.ac.in

**Pallab Dasgupta**
Department of Computer Science
Indian Institute of Technology Kharagpur
pallab@cse.iitkgp.ac.in

## 1 Appendix

### 1.1 Implementation Details

We use the following Feed Forward Neural Network architectures to train the original RL policies and the sub policies on different environments. The details of the actor network is as follows:
(layer1): Linear(in_features=observation_space, out_features=64, bias=True) activation : ReLU
(layer2): Linear(in_features=64, out_features=64, bias=True) activation : ReLU
(layer3): Linear(in_features=64, out_features=action_space, bias=True)
If the action space is discrete then a softmax activation is applied on layer 3. The details of the critic network is as follows:
(layer1): Linear(in_features=observation_space, out_features=64, bias=True) activation : ReLU
(layer2): Linear(in_features=64, out_features=64, bias=True) activation : ReLU
(layer3): Linear(in_features=64, out_features=1, bias=True)
Hyper-parameters are reported in Table 1. The Hyper parameters for training the original policy are chosen as per RL ppo baselines which are available in [2]. Reducing the learning rate during sub-policy training shows faster convergence over less number of trajectories. Open AI gym environments used for our experiments has MIT Licence which permits unrestricted use. Figure 4 shows the weight distribution of the original policy and the updated policy for different gym environments.

| Hyper-parameter | Value |
|---|---|
| Horizon (T) | 2048 |
| Adam step-size (Policy) | (1,2.5,3)e-4 |
| Adam step-size (Sub-Policy) | (1,2.5,3)e-3 |
| Num. epochs | 10 |
| Minibatch size | 64 |
| clip | 0.2 |
| Discount ($\gamma$) | 0.99 |
| GAE ($\lambda$) | 0.95 |
| Importance of objective function value ($\beta$) | 10 |

Table 1: PPO hyper-parameters used for training the policies and sub-policies for RL agents in open AI gym environments.

---

[*]Code is available online at https://github.com/britig/policy-refinement-bo

[2]https://github.com/araffin/rl-baselines-zoo/blob/master/hyperparams/ppo2.yml

35th Conference on Neural Information Processing Systems (NeurIPS 2021).

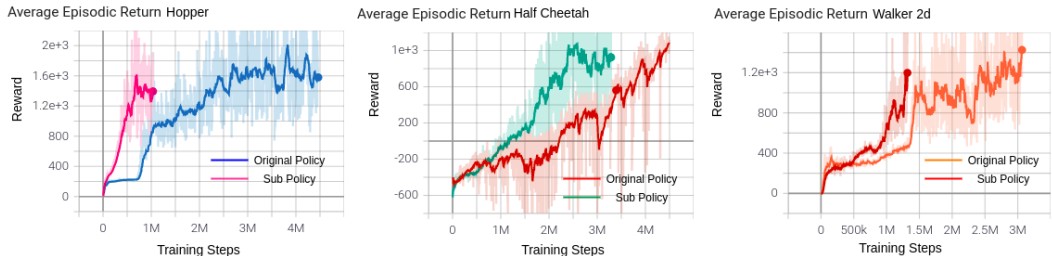

Figure 1: Reward plot for policies trained on MuJoCo environments showing training steps and the reward obtained by the sub-policy and the original policy

Table 2: Description of assertions tested on PPO policy learnt for MuJoCo environments along with the parameters considered as domain of uncertainty. We also report the number of failure trajectories corrected for each policy and the variation distance between the updated policy and the original policy. In our experiments n=1000

| Environment | Assertions | Parameter Bounds | Failure Trajecto-ries | Distance |
|---|---|---|---|---|
| Hopper-v2 | 1. Number of steps the hopper can take being vertical > 500 | Position : [(-0.05, 0.05)] * 6 Velocity : [(-0.05, 0.05)] * 6 | 22.6 ± 3.97 | 5.937 ± 1.053 |
| HalfCheetah-v2 | 1. Reward > 500 | Position : [(-0.1, 0.1)] * 9 | 16.6 ± 2.70 | 5.356 ± 1.224 |
| Walker2d-v2 | 1. Number of steps the walker can take being vertical > 300 | Position : [(-0.05, 0.05)] * 9 | 36.4 ± 8.73 | 5.818 ± 1.475 |

## 1.2 Experiments on MuJoCo Environment

We run additional experiments on MuJoCo Environments available in OpenAI Gym. MuJoCo is a fast physics based simulator for continuous control tasks. MuJoCo simulator has a Roboti LLC licence. We take three control tasks as mentioned in Table 2. The tasks primarily deal with teaching different types of robots to walk. We test the policies against similar assertions which states that robot should not fall before a particular amount of iteration. The domains of uncertainty is taken to be the position or velocity of the robot or both. The number of failure trajectories is reported in Table 2. The reward plots of the original policy and sub-policy are shown in Figure 1.

## 1.3 Additional Details on Bayesian Optimization

*Finding multiple counterexamples:* Multiple counterexamples can be obtained by eliminating the counterexample (minima) already explored from the search space. The counterexamples are (by definition) of negative sign ($\mu_k < 0$) where $\mu_k \in \varphi$. Therefore, if we square the function being optimized, $\mu_k$, then all its minima (negative function valuations) become positive and rise above zero. The points originally at zero remain at zero. These zero points serve as minima for the squared function. We find the nearest zeros surrounding the counterexample in each parametric direction and eliminate the intermediate points (namely, the valley containing the counterexample). For example, In a function having 2 variables let us assume the minimum point has been found at $(x_m, y_m)$. The nearest zero points for x dimension are $(x_{zl}, x_{zr})$ and y dimension are $(y_{zl}, y_{zr})$ then the region for search is illustrated in Fig 2a. The hyper-rectangular abstraction over the minima valley that is removed from the search space is shown in Figure 2b. This elimination is only carried out when the environment seed is fixed. When the environment is randomly changed as per seed same parameters could cause failure in a different environment hence we do not eliminate the already explored parameter. We wish to point out that the proposed principle of counter-example guided policy refinement using gradient updates remains the same regardless of the source of the counter-examples, that is, whether they are discovered using random sampling or using Bayesian optimization.

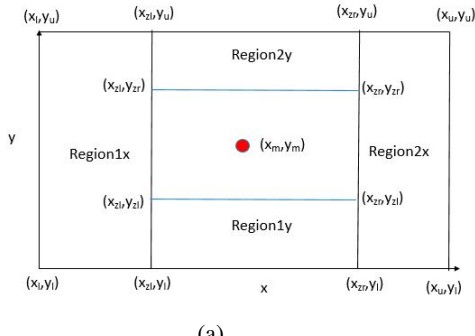

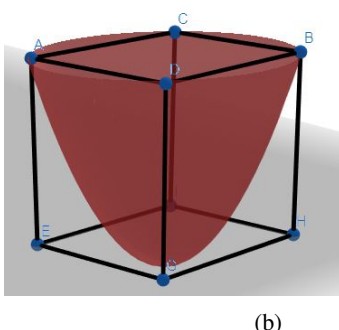

(a)                  (b)

Figure 2: a) New search region construction for two dimensions. Four new regions are constructed with 2 for x dimension and 2 for y dimension b) Rectangular abstraction around a minima valley considering nearest zero points of a function

We present table 3 on comparing random search and grid search with Bayesian Optimization on different environments. We executed these methods for 20 iterations each having 200 testing samples. We report the mean and standard deviation of the number of counterexamples discovered.

Table 3: Comparison of Bayesian Optimization for finding counterexamples with other search methods. Each method is executed for 20 iterations each having 200 testing samples

| Environment | Random Search | Grid Search | Bayesian Optimization |
|---|---|---|---|
| Pendulum-v0 | $2 \pm 2.41$ | 11 | $80.1 \pm 1.85$ |
| Bipedal-Walker-v3 | $35.6 \pm 6.58$ | 39 | $40.6 \pm 4.08$ |
| LunarLander-Continuous-v2 | $1.7 \pm 2.67$ | 3 | $40.85 \pm 5.14$ |
| Cart-pole-v0 | $123.7 \pm 5.33$ | 159 | $174.4 \pm 0.51$ |

For the pendulum, lunar-lander environments the number of counterexamples discovered is very low as random search and grid search do not target their search towards samples that had already violated the optimization function. In contrast to random search, Bayesian Optimization samples the next parameter value in an informed way to spend more time evaluating promising values. Grid search is computationally expensive being an exhaustive method and heavily dependent on the step the size chosen for the grid construction. In conclusion, Bayesian Optimization leads to fewer evaluations of the objective function and the generation of more counterexamples compared to random or grid search. It may be possible that biasing the random sampling in some way improves the effectiveness of random sampling for these experiments, but that is not the focus of this paper.

## 1.4 Additional Details

*Example 3.1* A simple neural network with 3 layers and the following architecture
(0): Linear(in_features=2, out_features=4, bias=True) (1): LeakyReLU(negative_slope=0.01)
(2): Linear(in_features=4, out_features=4, bias=True) (3): LeakyReLU(negative_slope=0.01)
(4): Linear(in_features=4, out_features=2, bias=True) (5): Softmax(dim=-1)
is trained using policy gradients to learn the policy for example 3.1. The input to the network is the observed state and output is the probability of taking an action. The weights of layer 1 of the network before (with violation of the safety specification) and after update are as follows:

$$\begin{bmatrix} 0.43851388 & -0.55869913 \\ 0.5483277 & 0.48225728 \\ 0.5648445 & -0.5123989 \\ -0.5019785 & 0.3065937 \end{bmatrix} \begin{bmatrix} 0.43711564 & -0.5587921 \\ 0.606144 & 0.48241794 \\ 0.6240289 & -0.5121144 \\ -0.4448014 & 0.306742 \end{bmatrix}$$

Where the first weight matrix corresponds to the original policy and the second weight matrix correspond to the updated policy. The weight matrix for layer 2 of the neural network for example

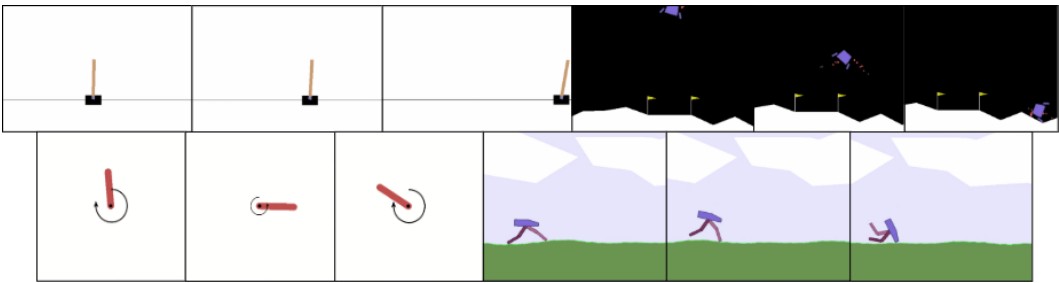

Figure 3: Failure trajectories on Cartpole, Lunar Lander, Pendulum and Bipedal Walker environments

3.1 is as follows:

$$\begin{bmatrix} -0.45758674 & 0.18934196 & 0.18558824 & 0.10932109 \\ 0.51084197 & 0.12021314 & -0.41461787 & 0.39017412 \\ 0.19967748 & -0.26843205 & -0.429302 & 0.25509173 \\ 0.44522664 & -0.37972566 & -0.00260387 & 0.20933793 \end{bmatrix}$$

$$\begin{bmatrix} -0.4576021 & 0.18934736 & 0.24492319 & 0.10932587 \\ 0.5109593 & 0.12012553 & -0.4133912 & 0.39011863 \\ 0.19976875 & -0.26851338 & -0.48902166 & 0.25499403 \\ 0.44514257 & -0.3796423 & 0.05719553 & 0.20934303 \end{bmatrix}$$

And layer 3 is as follows:

$$\begin{bmatrix} -0.48531273 & -0.04153176 & 0.2852086 & 0.27305838 \\ 0.49918333 & -0.40918258 & -0.05026673 & 0.45694318 \end{bmatrix}$$

$$\begin{bmatrix} -0.4856912 & -0.0415807 & 0.32630202 & 0.27319628 \\ 0.4995618 & -0.40913367 & -0.09136011 & 0.45680526 \end{bmatrix}$$

This example shows that the learnt weights of a policy can be minimally modified using counterexample trajectories to correct the trajectories causing failure while retaining the good behaviour learnt from optimizing the reward function.

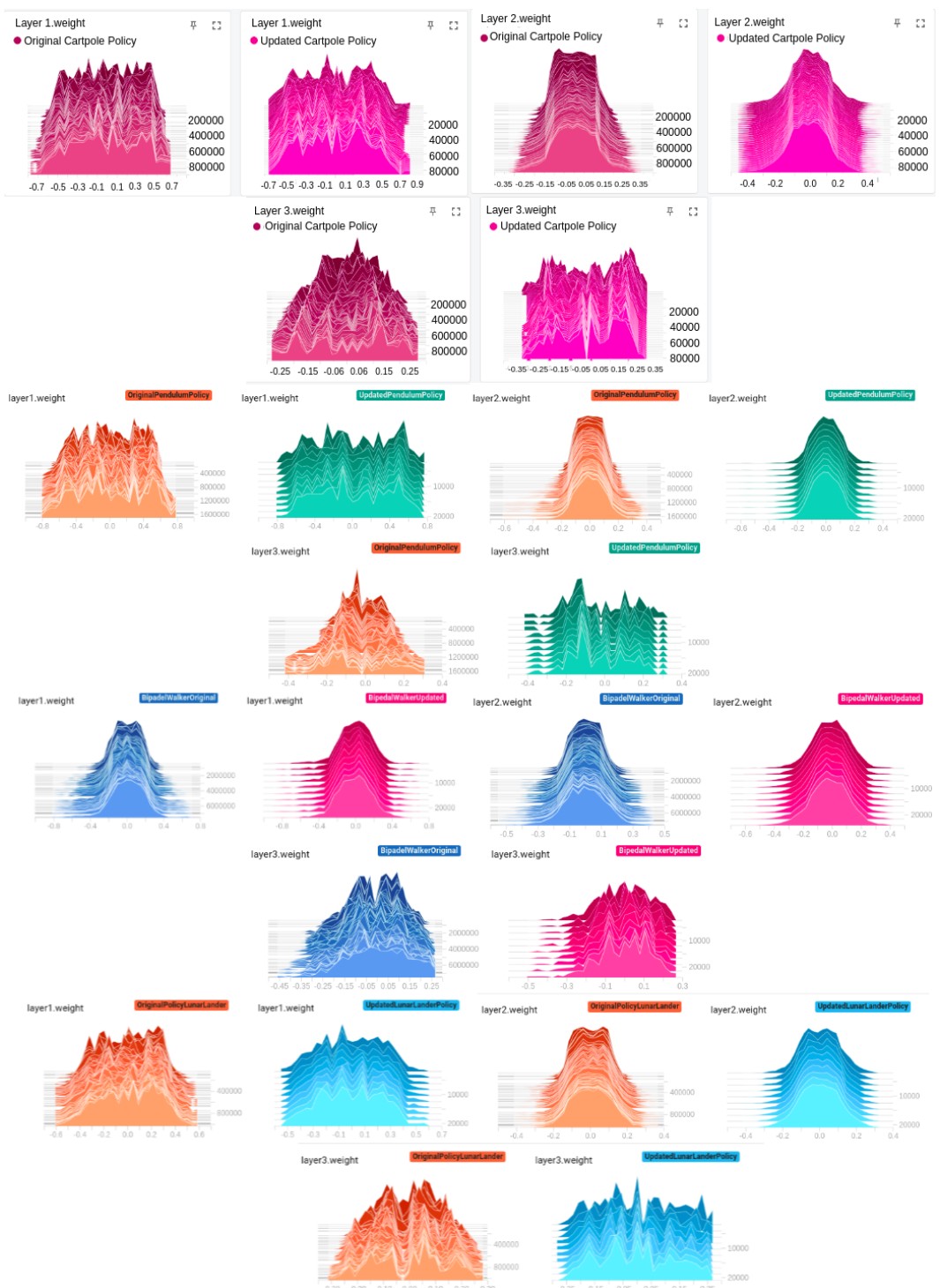

Figure 4: Weight distribution over multiple iterations of gradient update of the original policy and the updated policy for Cart-pole, Pendulum, Bipedal-Walker and Lunar Lander environment