# OpenReview forum: "Counterexample Guided RL Policy Refinement Using Bayesian Optimization"
_NeurIPS.cc/2021/Conference — NeurIPS 2021 Poster_

### Official Review · Reviewer_PvUR · 2021-07-06

**Rating:** 7
**Confidence:** 2

**Summary:**

The paper proposes a training methodology to incorporate constraints into a trained RL policy. It uses Bayesian optimization to select parameters for the MDP that are most likely to produce trajectories violating the constraints and then performs PPO updates to steer the policy away from these trajectories while remaining close to the original one. The experiments are conducted on popular OpenAI gym environments.

**Main Review:**

I'm not an expert on safe RL and I found the presentation of the topic given in the paper rather confusing. In particular it's not clear to me what exactly is the specific problem with directly penalizing constraint violation in the reward function that the authors are trying to solve. The authors claim that their method avoids such an approach, yet the proposed PPO updates look very much like adding negative reward for constraint violation. Those updates also seem to be largely perpendicular to the choice of using BO for MDP parameter selection and it's unclear to me whether the authors claim each of those separately as novel contributions or only their combination.

For me the fatal flaw of this paper is the underwhelming experimental evaluation, in particular not including any baselines. I don't quite understand how the original policy in the experiments was obtained, but my impression is that it was trained without any constraint violation penalties and I would very much like to see a comparison with a policy being trained in such a setting with normal PPO. On the BO side, at the very least we should see a comparison with random sampling. Without those experiments, I don't think the paper can be accepted.

EDIT:
raised score after rebuttal, see the rationale in the discussion below

**Time Spent Reviewing:**

2

---

> ### Author Response · Authors · 2021-08-09
> **Response to reviewer PvUR**
>
> Thanks for reviewing our paper. We hope the following elucidation changes your opinion on our contribution:
>
> 1) Bayesian Optimization vs Random Sampling: Comparison of these two methods have been reported previously in the literature, and it has been empirically shown that Bayesian optimization outperforms random sampling. The reviewer may refer to Figure 5 of reference [18] which shows BO is able to uncover more counterexamples than random sampling. BO has also been shown to perform better than other black box algorithms like simulated annealing, CMA-ES for test case identification for complex controllers. The reviewer may refer to Table 1 of reference [13]. We have therefore not presented any comparative study in this matter.
>
> 2) Lack of comparison with baselines: In the paragraph starting from Line 31, we have indicated that the contribution of the paper is a method for verifying and correcting a policy that has already been optimized and calibrated. Therefore, our method does not compare with existing methods for learning a safe policy from scratch. Calibrated and tested policies may fail after deployment because the real environment may present atypical situations that were not encountered during training. Such failures require a methodology for refining the pre-optimized policy in a targeted way, specifically with respect to the properties that are vulnerable in the deployed setting, while maintaining that the side effects on the policy are minor. We did not find any existing baseline methodology that addresses such a targeted refinement problem.
> However, for the sake of clarity, we provide the following experimental results. For comparison with our methodology, we take the following three PPO policies:
>
> (A) A PPO policy trained from scratch whose training trajectories are evaluated against the given properties and a negative penalty is added in case of property violation,
>
> (B) A PPO policy trained from scratch with only counterexample traces and negative penalty after one iteration of testing with BO, which is the same as the sub-policy actor-network $\pi_c$ from our paper,
>
> (C) A PPO policy trained from scratch with original training traces, counterexample traces and negative penalty after testing with one iteration of Bayesian Optimization, and
>
> (D) The refined policy obtained through our proposed method after the updates $\pi_{new}$. The parameter bounds (reported in the main manuscript) and safety properties for testing remain the same for all four policies.
>
> |Environment|Policy A| Policy B|Policy C |Policy D|
> |-----------|-----------|-----------|-----------|-----------|
> |Pendulum-v0|No. of Counterexamples: 89 Training Steps: 1.6M| No. of Counterexamples: 102 Training Steps: 850K|No. of Counterexamples: 0 Training Steps: 1.8M| No. of Counterexamples: 0 Training and Update Steps: 850K (Training sub-policy) + 20K (Update of original policy)|
> |Bipedal-Walker-v3|No. of Counterexamples: 45 Training Steps: 7.5M| No. of Counterexamples: 145 Training Steps: 2.8M|No. of Counterexamples: 41 Training Steps: 8M|No. of Counterexamples: 0 Training and Update Steps: 2.8M (Training sub-policy) + 20K (Update of original policy)|
> |LunarLander-Continuous-v2|No. of Counterexamples: 42 Training Steps: 1.1M|No. of Counterexamples: 18 Training Steps: 400K|No. of Counterexamples: 5 Training Steps: 1.2M|No. of Counterexamples: 0 Training and Update Steps: 400K (Training sub-policy) + 20K (Update of original policy)|
> |Cart-pole-v0|No. of Counterexamples: 179 Training Steps: 900K|No. of Counterexamples: 52 Training Steps: 150K|No. of Counterexamples: 0 Training Steps: 1M|No. of Counterexamples: 0 Training and Update Steps: 150K (Training sub-policy) + 80K (Update of original policy)|
>
> As can be seen from the comparisons, a trained policy from scratch with a negative penalty has the same number of counterexamples as a vanilla PPO policy trained without a negative penalty reported in the manuscript. The reason for this is that even though the policy was penalized with respect to the satisfaction of the safety property, the variable variations which occurred during deployment were not known during training. Policy B has counterexamples that had not occurred in the original policy as it was only trained on counterexample traces and did not retain the optimized behaviour of the original policy. In some cases, Policy C has fewer counterexamples than Policy A for the same tests but since it is trained again from scratch after BO testing, the training time is substantially higher than our policy refinement method. There are also cases (bipedal walker) where Policy C has a high number of counterexamples even after the inclusion of counterexample traces and negative penalty, as the updates are not targeted, exactly why an extensive verification becomes necessary. These experiments indicate that reward shaping is not a good refinement strategy. The refined policy by our methodology did not report any counterexamples for the same property even after 200 BO iterations and works with much fewer training and update steps. We shall add these results in our manuscript or supplement material if required.
>
> 3) Reward shaping versus PPO update: We do not change the reward function of the original policy because that would require retraining the network with the new reward function, with potential side effects on parts of the policy that are not sensitive to the safety property. On the other hand, we train a sub policy network $\pi_c$ based on the counterexamples discovered by Bayesian optimization and use that network to influence the choice of actions exactly in the relevant places (which we call saddle points) by making local policy gradient updates with PPO. We use failure penalties to train the sub policy actor $\pi_c$, but do not allow it to influence the reward function for the original policy $\pi_{old}$.
> The purpose of the PPO update is to do action shaping in targeted areas of the original pre-optimized and tested policy $\pi_{old}$. When the sub policy actor-network is used for updating the main policy to $\pi_{new}$, it changes the action probability (forcing the policy to learn correct action with advantage 1 and gradient update) for a given state. Hence, the policy is not trained from scratch with a new reward which is required in the case of reward shaping. The original policies are baseline policies obtained from trained PPO baselines (https://github.com/araffin/rl-baselines-zoo) as has been mentioned in the supplement material document. The combination of using BO to uncover failures and correcting the failures in a pre-optimized policy using gradient updates is the main contribution of our approach, and we believe that we are the first to discover the merits of this combination.

---

> > ### Comment · Reviewer_PvUR · 2021-08-18
> > **I'm convinced**
> >
> > I'm satisfied with the additional comparisons provided, I think that covers what I would regard as reasonable baselines and demonstrates a clear advantage of the proposed method. I would definitely like to see the table from the comment above included in the paper or at least the supplement. Since BO is used to generate counterexamples for refinement, it would be better to use random search to find safety violations in the final policies in order to avoid any potential confounding, but given the magnitude of improvement I expect the results to hold up qualitatively.
> >
> > I checked the references provided and in my opinion they provide the required evidence for why BO is being used instead of random sampling. I think this remark should be included in the paper for the benefit of future readers questioning this choice. That being said, if this is an established technique, this should be made clear in the abstract, which in its current form suggests to me that generating counterexamples with BO is being claimed as a novel contribution. Unless the proposed work improves that aspect as well, but I didn't get that impression.
> >
> > With the clarifications and additional results provided, I'm happy to recommend the acceptance of this paper, since it doesn't seem like the other reviewers objected to claims about novelty. I'll raise my score accordingly.

---

### Official Review · Reviewer_528P · 2021-07-13

**Rating:** 6
**Confidence:** 3

**Summary:**

this paper studies the problem of how to train a policy \pi which maximize rewards while retaining certain safety properties, stated as logical constraints. a BO algorithm discovers failure trajectories, which are then added to the PPO learning algorithm to steer the policy away from constraint violations.

**Limitations And Societal Impact:**

I do not see a "limitation and negative impact" section in the paper.

I find the check-list item "Did you discuss any potential negative societal impacts of your work? [N/A] paper presents works on safe RL with no foreseeable direct societal implications" incorrect, as safety in RL is (at least it is touted to be) very important for societal implications as we need to trust the AI we built. In fact, the first paragraph of the intro makes a very strong case that this line of work has huge safety implications. This is _inconsistent_ with how the paper handled the checklist item with [N/A].

**Main Review:**

### Originality and Significance:
I am a fan of this line of research, I think incorporating constraints as a way of specifying safety properties might be more clear to end-users than reward engineering or preference learning. As constraints are something (presumably) easy to understand by both algorithms and humans, it is a natural avenue to explore in human-machine communications in the realm of safety specifications.

### Clarity:
I do not think this paper is clearly written. Specifically, I spent a long time trying to understand what exactly is the query space for the BO.

To put it in lunar lander terms, is BO trying to discover a failure trajectory by:
1. put the lander into a weird initial location (x,y) with an odd angle so that it fails?
2. varying the mass / size of the lunar lander so that it fails?
3. forcing the policy \pi to make a low probability choice so it is more likely to fail?

The key phrase "Parameter Bound" seems to denote the space of query points for the BO, but is present in the paper only twice, and later in table1 there's a column named "domain of uncertainty". Are they the same thing? I'm assuming they are, but I am very lost.

I would suggest use lunar lander as the motivating example, as example 3.1 is way too detached from the main body of work to be the "opener example". I would also make the space of BO very explicit, if my understanding is correct, I would love to see a phrase of the form "randomly sampling mass and size of the lunar lander is unlikely to uncover a failure trajectory, but with BO we can actively query for a failure case of mass and size, and reliably sample failure trajectory with high probability. This is assuming the space of BO in lunar lander is this "parameter bound" or "domain of uncertainty", which could be wrong.

### Quality:
[edit : with the added random sampling baseline I think it is sufficient quality]

This work is (not yet) solid.

It is unclear to me from the experiments that BO is warranted for the domain in question. There should be some kind of metric showcasing how BO out performs just random sampling

It is also unclear to me that if the proposed method "proves" that no failure can occur, as the BO is working over a stochastic policy \pi, it may be possible that under some roll-outs, the policy will exhibit unsafe behaviour but simply "got lucky". At the very least, there should be an experiment that empirically justifies that your proposed algorithm gives a policy that is safer over many roll-outs.

I would suggest inclusion of a comprehensive table where, for each task (lander, walker, etc) you train three different policies:
\pi_a) without safety corrections, only maximizing rewards
\pi_b) with corrections, but sampled randomly without BO
\pi_c) with corrections, and sampled with BO
Then, evaluate how \pi_a, \pi_b, \pi_c performs on 10k new roll outs, notice the percentage of states that have failures. Hopefully, you'll discover that \pi_c have a very low number of failure states (this isn't going to be 0, due to the stochastic nature of \pi).

Overall, this work does not "prove" the resulting policy \pi_c is absolutely safe, and does not justify it empirically either.

**Time Spent Reviewing:**

1.5 hours

---

> ### Author Response · Authors · 2021-08-09
> **Response to reviewer 528P**
>
> We thank you for your comments. We address each of the points as below:
> 1) Bayesian Optimization Query Space: You are right, the domain of uncertainty stated in the paper is indeed captured by the parameter bounds. We shall clarify this in the manuscript. The parameters for carrying out Bayesian optimization (BO) are chosen from the backward cone of influence of the variables in the targeted safety property. The exact choice is typically an engineering decision based on the trigger for the refinement approach, which could be quite varied, such as encountering a failure of a safety property, or violation in one or more environment assumptions. The chosen parameters may include dynamic ones like limits of velocity and angle of tilt, as well as static system parameters such as mass and density. Mathematically, BO treats all of these as variables and aims to make the property evaluation function negative (in our setting) by sampling intelligently from the parameter ranges of the chosen variables. The sampled parameters are evaluated on the policy and based on these evaluations, BO samples new parameter values. This continues until the objective function evaluates to negative (that is, failure is encountered) or the time budget is exhausted. In our experiments, we consider the position and velocity of the lander as testing parameters for BO. The approach is used to find multiple counter-examples and train the sub policy network, which becomes the basis for policy gradient updates carried out in the second phase of our policy refinement approach.
>
> 2) Bayesian Optimization vs Random Sampling: Comparison of these two methods have been reported previously in the literature, and it has been empirically shown that Bayesian optimization outperforms random sampling. The reviewer may refer to Figure 5 of reference [18] which shows BO is able to uncover more counterexamples than random sampling. BO has also been shown to perform better than other black box algorithms like simulated annealing, CMA-ES for test case identification for complex controllers. The reviewer may refer to Table 1 of reference [13]. We have therefore not presented any comparative study in this matter.
> 3) Example 3.1: We prefer this example as it is visually simplistic to explain trajectories and their corrections using the simple grid world. It is difficult to explain trajectories on the lunar lander environment due to its continuous state space.
> 4) We have not claimed that failures can never happen with the updated policy. In practical settings, differences between the training environment and the deployed environment are almost ubiquitous because the RL agent looks at a subset of the variables defining the system and its environment. Consequently, absolute safety is rare. On the other hand, when an optimized and tested policy encounters a failure in the deployed environment, we may need to fix it in a targeted way and ensure that similar failures are not encountered again. Our experiments show that BO is unable to find new counterexamples for the target property after the policy is updated based on our approach. This shows that (A) the counterexamples found by BO before the policy update has been eliminated, and (B) the update did not result in new counterexamples for the same property in the same parameter space.
> In summary, our aim is to refine the policy locally based on observed counterexamples, and our approach successfully does this on a wide variety of test cases, including the ones provided in the supplementary material. This does not rule out failures when the parameter space for the deployed environment changes or when we target a new safety property.
> We would also like to emphasize that BO is working on a trained and tested policy, that is, this is not a methodology for learning a safe policy from scratch. Even though the training procedure is not deterministic and considers a multivariate normal distribution or categorical distribution over the actions, once the training has been completed, then the internal workings of a neural network are deterministic with fixed parameters. As per our implementation, during testing, we simply choose the action suggested by the policy network deterministically. This means that for a particular rollout in the test environment the neural network will suggest the same action and it will always display unsafe behaviours for a particular state if the counterexample is found. We show that the updated policy is safer by running the same BO test for 200 iterations with respect to the property and showing that no counterexamples are reported. This can also be validated by running our code provided as a part of the supplementary material.
> In the lines of the experiment suggested we provide the following comparisons with three different PPO policies:
>
> (A) A PPO policy trained from scratch whose training trajectories are evaluated against the given properties and a negative penalty is added in case of property violation,
>
> (B) A PPO policy trained from scratch with only counterexample traces and negative penalty after one iteration of testing with BO, which is the same as the sub-policy actor-network $\pi_c$ from our paper,
>
> (C) A PPO policy trained from scratch with original training traces, counterexample traces and negative penalty after testing with one iteration of Bayesian Optimization, and
>
> (D) The refined policy obtained through our proposed method after the updates $\pi_{new}$. The parameter bounds (reported in the main manuscript) and safety properties for testing remain the same for all four policies.
>
> |Environment|Policy A| Policy B|Policy C |Policy D|
> |-----------|-----------|-----------|-----------|-----------|
> |Pendulum-v0|No. of Counterexamples: 89 Training Steps: 1.6M| No. of Counterexamples: 102 Training Steps: 850K|No. of Counterexamples: 0 Training Steps: 1.8M| No. of Counterexamples: 0 Training and Update Steps: 850K (Training sub-policy) + 20K (Update of original policy)|
> |Bipedal-Walker-v3|No. of Counterexamples: 45 Training Steps: 7.5M| No. of Counterexamples: 145 Training Steps: 2.8M|No. of Counterexamples: 41 Training Steps: 8M|No. of Counterexamples: 0 Training and Update Steps: 2.8M (Training sub-policy) + 20K (Update of original policy)|
> |LunarLander-Continuous-v2|No. of Counterexamples: 42 Training Steps: 1.1M|No. of Counterexamples: 18 Training Steps: 400K|No. of Counterexamples: 5 Training Steps: 1.2M|No. of Counterexamples: 0 Training and Update Steps: 400K (Training sub-policy) + 20K (Update of original policy)|
> |Cart-pole-v0|No. of Counterexamples: 179 Training Steps: 900K|No. of Counterexamples: 52 Training Steps: 150K|No. of Counterexamples: 0 Training Steps: 1M|No. of Counterexamples: 0 Training and Update Steps: 150K (Training sub-policy) + 80K (Update of original policy)|
>
> As can be seen from the comparisons, a trained policy from scratch with a negative penalty has the same number of counterexamples as a vanilla PPO policy trained without a negative penalty reported in the manuscript. The reason for this is that even though the policy was penalized with respect to the satisfaction of the safety property, the variable variations which occurred during deployment were not known during training. Policy B has counterexamples that had not occurred in the original policy as it was only trained on counterexample traces and did not retain the optimized behaviour of the original policy. In some cases, Policy C has fewer counterexamples than Policy A for the same tests but since it is trained again from scratch after BO testing, the training time is substantially higher than our policy refinement method. There are also cases (bipedal walker) where Policy C has a high number of counterexamples even after the inclusion of counterexample traces and negative penalty, as the updates are not targeted, exactly why an extensive verification becomes necessary. These experiments indicate that reward shaping is not a good refinement strategy. The refined policy by our methodology did not report any counterexamples for the same property even after 200 BO iterations and works with much fewer training and update steps. We shall add these results in our manuscript or supplement material if required.
>
> 5) Kindly refer to lines 299 to 303 of our manuscript for our assessment of the limitations of the proposed approach.
>
> 6) Indeed, safety assurance in some systems have a significant societal impact. We shall add a section on this in the manuscript.

---

> > ### Comment · Reviewer_528P · 2021-08-13
> > **can you show me how random sampling does in comparison?**
> >
> > The references you provided (from what I can check) does not use the walker or the lander environment. Saying bayesian optimization works better for _other_ environments does not make it true in _your_ specific problem. It is like saying adding pepper makes food tastier in general, but it will not apply if you are making a cake. To truly _warrant_ your approach, like the other reviewers pointed out, this work needs to compare against some baselines. Random sampling is _the_ simplest baseline, but from personal experience, random sampling is fairly good at gather information, often nearly as well as a clever method such as BO for a lot of active learning scenarios. Not comparing your approach against a random sampler indicates to me (implicitly) that your approach performs no better than random sampling.

---

> > > ### Author Response · Authors · 2021-08-14
> > > **Experiments on effectiveness of Random Search, Grid Search vs Bayesian Optimization**
> > >
> > > We wish to point out that the proposed principle of counter-example guided policy refinement using gradient updates remains the same regardless of the source of the counter-examples, that is, whether they are discovered using random sampling or using Bayesian optimization.
> > > Nevertheless, in response to your observations, please find our report on comparing random search and grid search with Bayesian Optimization on different environments. We executed these methods for 20 iterations each having 200 testing samples. We report the mean and standard deviation of the number of counterexamples discovered. We may add these results as supplementary material.
> > >
> > > | Environments| Random Search |Grid Search|Bayesian Optimization |
> > > | ----------- | ----------- |----------- |----------- |
> > > |Pendulum-v0      | 2$\pm$2.41|11 |80.1$\pm$1.85|
> > > |Bipedal-Walker-v3| 35.6$\pm$6.58|39|40.6$\pm$4.08|
> > > |LunarLander-Continuous-v2 |1.7$\pm$2.67 |3 |40.85$\pm$5.14|
> > > |Cart-pole-v0|123.7$\pm$5.33|159|174.4$\pm$0.51|
> > >
> > > For the pendulum, lunar-lander environments the number of counterexamples discovered is very low as random search and grid search do not target their search towards samples that had already violated the optimization function. In contrast to random search, Bayesian Optimization samples the next parameter value in an informed way to spend more time evaluating promising values. Grid search is computationally expensive being an exhaustive method and heavily dependent on the step the size chosen for the grid construction. In conclusion, Bayesian Optimization leads to fewer evaluations of the objective function and the generation of more counterexamples compared to random or grid search.
> > >
> > > It may be possible that biasing the random sampling in some way improves the effectiveness of random sampling for these experiments, but that is not the focus of this paper.

---

> > > > ### Comment · Reviewer_528P · 2021-08-18
> > > > **thanks for this**
> > > >
> > > > really this is all I needed. raising score to 6 as that was the biggest issue I had.
> > > >
> > > > I agree with pvur's point on this "Since BO is used to generate counterexamples for refinement, it would be better to use random search to find safety violations in the final policies in order to avoid any potential confounding, but given the magnitude of improvement I expect the results to hold up qualitatively" as well, and would advocate to use strict random samples (a lot more of them if needed) to _evaluate_ the final policies as well as BO's biased sampling.

---

### Official Review · Reviewer_4nks · 2021-07-17

**Rating:** 7
**Confidence:** 3

**Summary:**

This paper presents a counterexample-guided policy refinement method to check and correct an already trained RL policy. The proposed algorithm (i) uses Bayesian Optimization to find failure cases in a policy, (ii) then learns a new sub-policy to locally overcome the failed cases, and (iii) finally uses a slightly modified PPO algorithm to refine/update the initial policy to eliminate the failure cases in a proximal manner. The paper is evaluated on several OpenAI Gym environments and shows they are able to find and fix several failure cases with a small mean variation distance between the policies.


**Limitations And Societal Impact:**

Yes.

**Main Review:**

Strengths:
- Using a counterexample guided approach in RL to ensure safety is an interesting approach
- The algorithm to proximally refine the policy is neat and well done

Weakness:

This paper needs a more thorough evaluation including comparing to a few baselines, evaluating on more metrics, and on harder problems
- With the lack of baselines, it is hard to appreciate the numbers in the evaluation. One obvious baseline would be training/fine-tuning with the combined reward function (reward + loss penalty, the total reward used for training $\pi_c$) on all initial states (essentially eliminating the Bayesian Optimization step.
- Also, please report the initial reward, final reward, # failure cases in the final policy.
- It will also be interesting to see how this algorithm affects the sample complexity (which needs to take into account the sample complexity for both the Bayesian optimization and the retraining steps) and compare to other baselines/approaches' sample complexity.
- A tricky part of any counter-example guided algorithm is that it requires multiple back and forth iterations and sometimes it does not converge. When you fix a policy for some failure cases, more and newer failure cases creep up. However, in the current evaluation, it seems like all problems are solved with just 1 iteration. Can the authors comment on this? Is this some special property of your algorithm or are the tasks too simple?



**Time Spent Reviewing:**

2

---

> ### Author Response · Authors · 2021-08-09
> **Response to Reviewer 4nks**
>
> We thank you for your comments. Our response to each of your points are provided below:
> 1) Lack of comparison with baselines: In the paragraph starting from Line 31, we have indicated that the contribution of the paper is a method for verifying and correcting a policy that has already been optimized and calibrated. Therefore, our method does not compare with existing methods for learning a safe policy from scratch. Calibrated and tested policies may fail after deployment because the real environment may present atypical situations that were not encountered during training. Such failures require a methodology for refining the pre-optimized policy in a targeted way, specifically with respect to the properties that are vulnerable in the deployed setting, while maintaining that the side effects on the policy are minor. We did not find any existing baseline methodology that addresses such a targeted refinement problem.
> Your suggestion for a baseline that incorporates reward along with loss will require re-training the agent after each change in the combined reward function, and we will lose the ability to localize the policy updates to only those states that are responsible for the failure. As a consequence, the new policy will need to be reverted for testing and calibration, and this effort is typically very significant, but difficult to quantify.
> On the other hand, our methodology only does a targeted action shaping on the original policy by updating the gradient in favour of the correct actions. We use Bayesian Optimization to find the counterexamples targeting specific safety properties, and prepare the sub policy actor-network $\pi_c$ which is used to update the gradient for refining the original policy.
> Kindly see the supplementary material provided with our original submission for more experiments on harder environments.
> Although we reiterate that there seems to be no existing methodology for our problem, we understand your discomfort due to the absence of comparison. Therefore, we show some results to address your concern. We compare the following PPO policies:
>
> (A) A PPO policy trained from scratch whose training trajectories are evaluated against the given properties and a negative penalty is added in case of property violation,
>
> (B) A PPO policy trained from scratch with only counterexample traces and negative penalty after one iteration of testing with BO, which is the same as the sub-policy actor-network $\pi_c$ from our paper,
>
> (C) A PPO policy trained from scratch with original training traces, counterexample traces and negative penalty after testing with one iteration of Bayesian Optimization, and
>
> (D) The refined policy obtained through our proposed method after the updates $\pi_{new}$. The parameter bounds (reported in the main manuscript) and safety properties for testing remain the same for all four policies.
>
> |Environment|Policy A| Policy B|Policy C |Policy D|
> |-----------|-----------|-----------|-----------|-----------|
> |Pendulum-v0|No. of Counterexamples: 89 Training Steps: 1.6M| No. of Counterexamples: 102 Training Steps: 850K|No. of Counterexamples: 0 Training Steps: 1.8M| No. of Counterexamples: 0 Training and Update Steps: 850K (Training sub-policy) + 20K (Update of original policy)|
> |Bipedal-Walker-v3|No. of Counterexamples: 45 Training Steps: 7.5M| No. of Counterexamples: 145 Training Steps: 2.8M|No. of Counterexamples: 41 Training Steps: 8M|No. of Counterexamples: 0 Training and Update Steps: 2.8M (Training sub-policy) + 20K (Update of original policy)|
> |LunarLander-Continuous-v2|No. of Counterexamples: 42 Training Steps: 1.1M|No. of Counterexamples: 18 Training Steps: 400K|No. of Counterexamples: 5 Training Steps: 1.2M|No. of Counterexamples: 0 Training and Update Steps: 400K (Training sub-policy) + 20K (Update of original policy)|
> |Cart-pole-v0|No. of Counterexamples: 179 Training Steps: 900K|No. of Counterexamples: 52 Training Steps: 150K|No. of Counterexamples: 0 Training Steps: 1M|No. of Counterexamples: 0 Training and Update Steps: 150K (Training sub-policy) + 80K (Update of original policy)|
>
> As can be seen from the comparisons, a trained policy from scratch with a negative penalty has the same number of counterexamples as a vanilla PPO policy trained without a negative penalty reported in the manuscript. The reason for this is that even though the policy was penalized with respect to the satisfaction of the safety property, the variable variations which occurred during deployment were not known during training. Policy B has counterexamples that had not occurred in the original policy as it was only trained on counterexample traces and did not retain the optimized behaviour of the original policy. In some cases, Policy C has fewer counterexamples than Policy A for the same tests but since it is trained again from scratch after BO testing, the training time is substantially higher than our policy refinement method. There are also cases (bipedal walker) where Policy C has a high number of counterexamples even after the inclusion of counterexample traces and negative penalty, as the updates are not targeted, exactly why an extensive verification becomes necessary. These experiments indicate that reward shaping is not a good refinement strategy. The refined policy by our methodology did not report any counterexamples for the same property even after 200 BO iterations and works with much fewer training and update steps. We shall add these results in our manuscript or supplement material if required.
>
> 2) In the proposed methodology for policy refinement, the reward updates take place only in the sub policy network $\pi_c$ (that is, for sub policy training), and is based on the counterexample trajectories uncovered by Bayesian optimization. The reward/penalty of a trajectory depends on the margins by which the trajectory violates the property. In the Lunar Lander experiment, the reward for a successful run is 200. Consider Assertion 1, which specifies the position to be between -0.4 to 0.4. The total reward for a trajectory where the lander lands at 0.8 will be 200 + 50*(0.4 – 0.8) = 180, where 50 is the penalty per unit deviation. Likewise, the penalty for a trajectory that lands at 0.6 is 190. We reiterate that these rewards are used only in the sub policy network, and are not used for shaping the rewards of the original policy. Since the final reward is different for each counterexample trace it has not been reported in the paper.
>
> 3) The sample complexity for training the sub policy depends on the number of counterexamples found. The sample complexity of BO depends on the number of parameters. Neither of these requires a large number of samples in practice due to the targeted nature of the refinement. Theoretically, sample complexity is asymptotically the same as training an RL algorithm from scratch.
>
> 4) The refinement needs to consider multiple counter-examples to rule out variants of the counter-example scenario. Following the method prescribed in reference [15] in the manuscript, we are able to use Bayesian optimization (BO) to uncover multiple counterexamples in one cycle of testing (that is, we use BO repeatedly to uncover a set of counterexamples before proceeding to update the policy). This is also elaborated in the pdf provided with supplementary materials. However, as BO is a statistical method we run the BO tests multiple times within one iteration and take the highest number of counterexamples obtained for correction. As shown in Figure 3a, the sub-policy is trained over multiple iterations and the correction happens over multiple updates. The number of updates for policy correction for each task is given in the plots of Fig 3b in the main manuscript and Fig 4 in supplement material. The y axis shows the number of steps and the x-axis shows the weight distribution of the updated policy. In the presented cases, at the end of the first update phase, we tested the updated policy via BO for seeking more counterexamples. Even after 200 BO iterations, no new counterexamples were found. We believe that this happened because (A) the sub policy network was trained with multiple counterexamples uncovered by BO in the first cycle thereby covering the possible failure trajectories for the targeted property, and (B) the targeted action shaping did not result in side effects, and hence no new failures. Theoretically, it is possible to have cases where a single cycle will not suffice, but we did not find such cases in practice.

---

> > ### Comment · Reviewer_4nks · 2021-08-18
> > **Reply**
> >
> > Thank you for doing the new experiments. They increased my confidence and I am happy to increase my score. Also, please add these results to the main paper (not the appendix). The baselines are needed for a reader to properly understand the challenges in the different benchmarks.
> >
> > I also agree with the other reviewers to use a technique other than BO to evaluate the number of final failure cases.

---

### Official Review · Reviewer_yDKZ · 2021-07-17

**Rating:** 7
**Confidence:** 4

**Summary:**

Authors introduce a safety-oriented policy-refinement method for reinforcement learning. Authors use Bayesian optimization to find safety-violation traces from a policy generated by a vanilla RL algorithm. Those traces are latter use to produce a new trained policy that satisfies the requirements. The method is meant for continuous-action environments

**Limitations And Societal Impact:**

Authors explicitly state their limitations in the conclusions

**Main Review:**

Authors present a novel method for generating safe RL agents as far as I am aware, both the process of finding counterexamples through Bayesian optimization and  their policy refinement method are interesting contributions. However, authors claim two times that their are first to refine learned RL policies through counterexamples. This is not the case since [1] already did that. However the proposed method is significantly different from that previous approach. Moreover, the proposed method is applicable to a wider range of applications since it can handle larger state spaces than the model checking approach from [1].

The paper is clear and well written, authors introduce various images and running examples that makes the approach intuitive and easy to follow. They provide sufficient proof and explanation with definitions and various experimental benchmarks. I only saw a handful minor things that needed correction.

Overall is a good contribution and I recommend acceptance (upon that authors correct the two times they say to be first to present a counter-example refinement method for RL and include [1] in their references).

Additional comments:
- Error intervals are barely visible  in figure 3, I strongly suggest to draw the plots with a different tool than tensorboard
- What do the error intervals represent, quartiles? std?
- Table titles should be above the table according to neurips style, also in the caption, II suggest recalling what n is

[1] Zhou, W., & Li, W. (2018, July). Safety-aware apprenticeship learning. In International Conference on Computer Aided Verification (pp. 662-680). Springer, Cham.

**Time Spent Reviewing:**

6

---

> ### Author Response · Authors · 2021-08-09
> **Response to reviewer yDKZ**
>
> The reference [1] has not been provided in your review comments. Nevertheless, if you kindly identify this reference, we shall be happy to cite it and change the necessary sentences to disambiguate our contribution from that paper. You have indicated that our approach is very different from that of [1], so this should not be a problem.
> In line with the additional comments, we will work towards changing the presentation of Figure 3. The error intervals represent the standard deviation. We thank you for pointing out the formatting issues and we shall change them as per NeurIPS guidelines.
> Also, thank you for the appreciation !!

---

> > ### Comment · Reviewer_yDKZ · 2021-08-11
> > **Reference added**
> >
> > I am sorry for the omission, you can find the previous work now.

---

> > > ### Author Response · Authors · 2021-08-14
> > > **Reference updated in the manuscript**
> > >
> > > Thank you for identifying the reference. We have now cited this work in our manuscript and disambiguated our contributions.

---

### Author Response · Authors · 2021-08-09
**Response**

We thank all reviewers for their valuable feedback. We would like to first reiterate our main contribution and then respond to the individual reviews.

**Main Contribution**: Our method focuses on verifying and correcting a policy that has already been optimized and calibrated. We do not propose an algorithm that learns a safe policy from scratch rather refines a policy to become safer if counterexamples are found during testing. The testing is done using Bayesian Optimization with respect to parameter bounds on variables and a given property. Failures may occur in a policy that had been trained with negative penalties for constraint violation in testing due to changes in environment behaviour during deployment or due to scenarios that had not been encountered during training. Our aim is to fine-tune the trained policy with minimalistic change in order to tackle the counterexamples (bugs) that are found during testing.

---

### Decision · Program_Chairs · 2021-09-27

**Decision:**

Accept (Poster)

**Comment:**

This paper has considerable support from the reviewers but not does not seem to rise to the level of an oral.

For me the formal adversarial set up is strange.  Some form importance sampling under an "unsafe" state-transition bias would yield an estimate of the probability of a safety violation. Optimizing an estimate of the probability of a safety violation, possibly with stochastically drawn model parameters, seems like a cleaner conceptual set up to me.  But the reviewers are happy and I will go along.